# Analysis of Point Shear Wave Elastography and Biochemical Markers for the Detection of Liver Fibrosis

**DOI:** 10.3390/medicina57010040

**Published:** 2021-01-06

**Authors:** Maciej Cebula, Katarzyna Gruszczyńska, Marek Hartleb, Jan Baron

**Affiliations:** 1Department of Radiodiagnostics and Invasive Radiology, School of Medicine in Katowice, Medical University of Silesia, 40-752 Katowice, Poland; jbaron@sum.edu.pl; 2Department of Diagnostic Imaging, School of Medicine in Katowice, Medical University of Silesia, 40-752 Katowice, Poland; kgruszczynska@poczta.onet.pl; 3Department of Gastroenterology and Hepatology, School of Medicine in Katowice, Medical University of Silesia, 40-752 Katowice, Poland; mhartleb@sum.edu.pl

**Keywords:** elastography, liver, fibrosis, point shear wave elastography (pSWE)

## Abstract

*Background and Objectives:* This work focuses on the possibility of using the point shear wave elastography (pSWE) method for detecting biochemical markers in diffuse liver diseases. Additionally, this study addresses the issue of the influence of ultrasound factors on the pSWE quality indicators of the obtained measurements. *Materials and Methods:* A pSWE examination was performed on 139 patients (69 female and 70 male) diagnosed with diffuse liver disease. The average age for all patients was 50.7 ± 15.0 years (female: 52.7 ± 15.2 years; male: 48.8 ± 14.6 years). Of these 139 patients, 65 met the inclusion criteria regarding biochemical parameters. The pSWE quality indicators were related to abnormalities found in B-mode ultrasound. *Results:* A strong positive correlation was found between the results of the pSWE and all biochemical indexes analysed, with the exception of age/platelet count (PLT), for which an average correlation was obtained. The greatest correlation was observed between the elastography and King’s Score index. There was no correlation observed between elastography and any of the analysed parameters or biochemical indexes considered. The pSWE measurements were impaired by factors such as thick soft tissue, uneven hepatic surface, hepatomegaly and female gender. No statistically significant difference in pSWE quality indicators parameters was found between disease entities. *Conclusions:* pSWE seems to be a complementary method for detecting biochemical indexes, but its results can be influenced by numerous factors.

## 1. Introduction

Diffuse liver disease is difficult to diagnose because of its inert course and general symptoms. The detection of early stage chronic liver disease is still low despite the implementation of screening and social education. A shear wave elastography (SWE) is a new method used in the diagnosis of liver diseases. With an adequately equipped ultrasound machine, the elastography can be performed during a standard abdominal ultrasound. The cost efficiency, absence of ionizing radiation and minimal contraindications are the main perks of this method [1].

In order to obtain valid measurements, the proper patient preparation, lack of notable alterations in liver function tests and acquisition of predefined measurement quality indicators levels are required [2,3]. Unfortunately, even when the first two conditions are met, obtaining results has proven difficult or even impossible in some patients.

The main goal of the study is to compare the biochemical indices with SWE results in search of the best matching one. The secondary aims of the study are to search for the cofounders of SWE measurement visible in a B-mode ultrasound and to look for diseases with increased risk of SWE measurement error.

## 2. Materials and Methods

This study is a retrospective analysis of point shear wave elastography (pSWE) examinations, which were performed at the Samodzielna Pracownia Diagnostyki Obrazowej Tomograf Sp. z o.o., Tychy, Poland and the Department of Radiodiagnostics and Interventional Radiology of the Central Clinical Hospital, Katowice, Poland. The Arietta V70 Hitachi Aloca devices (Berkshire, UK) with a convex C251 (1.8–5.0 MHz) probes were used. All examinations were performed in accordance with the updated European Federation of Societies for Ultrasound in Medicine and Biology guidelines and Polish Ultrasound Society guidelines [3,4].

A local ethics committee, the Komisja Bioetyczna Śląskiego Uniwersytetu Medycznego w Katowicach, waived the requirement to obtain ethical approval for this study. Patient records were obtained in the form of anonymised reports, and identification of individual patients solely based on the collected data was impossible.

The patients with diffuse liver disease, initially diagnosed by the referring physician, were included in the study group. For the purpose of this work, the definition of diffuse liver disease included autoimmune hepatitis, hepatitis B and C, non-alcoholic fatty liver disease (NAFLD), primary biliary cholangitis, primary sclerosing cholangitis, idiopathic cirrhosis and idiopathic elevation of the alanine transaminase (ALT) or aspartate aminotransferase (AST). Patients who did not meet preparation for pSWE examination criteria were not enrolled. In patients with arterial hypertension or diabetes, an additional precondition of stabilized blood pressure or glucose levels, respectively, was required [5]. The AST, ALT, international normalised ratio (INR) and platelet count (PLT) parameters were counted and recorded ±7 days before the start of the pSWE examination in order to estimate the liver fibrosis severity.

Variables such as age, gender, liver size, liver hyperechogenicity, soft tissue thickness (distance between probe surface and liver capsule), hepatic capsule unevenness, high liver orientation, percentage of effective vs. efficiency rate (VsN, reliability indicator that allows to test the measurement validity) and interquartile range of vs. group to median ratio (iQR/M) were also noted.

The cut-out values of quality indicators of SWE measurement were VsN > 60%, iQR/M < 30 % for elasticity and iQR/M < 15% for SWE velocity measurements.

On the basis of gathered biochemistry information, the following indices were calculated according to the formulas provided.

The De Ritis ratio is calculated using: $$\mathrm{AST}/\mathrm{ALT}=\;\frac{\mathrm{AST}\;\left(U/L\right)}{\mathrm{ALT}\;\left(U/L\right)}\;$$
where AST is the aspartate aminotransferase level, and ALT is the alanine transaminase level. Table 1 presents the point ranges for the age/PLT index, that is a sum of points from presented columns.

The AST to platelet ratio index (APRI) is found by: $$\mathrm{APRI}=\;\frac{\mathrm{AST}\left(/\mathrm{ULN}\right)\times 100}{\mathrm{PLT}\;}$$
where AST (/ULN) is the aspartate aminotransferase to the upper limit of the normal value ratio. The LOK index is calculated using:$$\mathrm{LOK}\;\mathrm{index}=\;\frac{e^{\left(1.26\times \frac{\mathrm{AST}}{\mathrm{ALT}}\right)+\left(5.27\times \mathrm{INR}\right)-\left(0.0089\times \mathrm{PLT}\right)-5.56}}{1+e^{\left(1.26\times \frac{\mathrm{AST}}{\mathrm{ALT}}\right)+\left(5.27\times \mathrm{INR}\right)-\left(0.0089\times \mathrm{PLT}\right)-5.56}}$$
where INR is the international normalised ratio. To determine the Fibrosis-4 score (FIB-4), the following equation is used:$$\mathrm{FIB}-4=\;\frac{\mathrm{Age}\times \mathrm{AST}}{\mathrm{PLT}\times \;\sqrt{\mathrm{ALT}}}$$

King’s score is calculated using:$$\mathrm{King}’\;\mathrm{score}=\;\frac{\mathrm{Age}\times \mathrm{AST}\times \mathrm{INR}}{\mathrm{PLT}}$$

Finally, the Goteborg University cirrhosis index (GUCI) can be found by:$$\mathrm{GUCI}=\;\frac{\mathrm{AST}\;\left(/\mathrm{ULN}\right)\times \mathrm{INR}\times 100}{\mathrm{PLT}\;}$$

In order to determine the best match of pSWE and the biochemical indices, the elasticity, shear wave propagation velocity measurement (SWM1-Vs) and biochemical indices values correlation strength were compared. To build a model that would maximally predict liver stiffness and SWM1-Vs based on biochemical parameters, progressive and backward regression were used. Finally, residual analysis was performed, and the model remained statistically significant before and after elimination of the outliers.

After that, an exploratory statistical analysis was performed. Correlating factors in the group of patients for whom correct parameters were not achieved (after excluding known causes) were explored.

A number of point shear wave speed measurements (PSWSMs), needed to obtain a sound result of the entire examination, were compared in search of difficult-to-evaluate liver pathologies.

The assessment of the data distribution was made by assessing the collected values using both descriptive statistics and visual analysis of the charts. The normality of the distribution of variables was checked using the Kolomogorov-Smirnov, Lilliefors and Shapiro-Wilk tests. In the case of normal distribution of groups and homogeneous variances, the F test was used together with post-hoc tests to compare groups of patients based on multi-state variables. In the absence of one of these conditions, the analysis was carried out using the Kruskal-Wallis and post-hoc tests. In order to compare the collected results against binary variables, Student’s *t*-test was used if the criteria were met, or the Mann-Whitney U test was used if they were not met. For the correlation analysis, Spearman’s R test was used.

## 3. Results

The group of 139 patients (69 female and 70 male) were evaluated by pSWE. The average age of the entire patient group was 50.73 ± 14.99 years (female: 52.72 ± 15.22 years; male: 48.77 ± 14.60 years). Of the 139 patients, just 67 met the biochemical parameters inclusion criteria. One case each of hemochromatosis and benign recurrent intrahepatic cholestasis were excluded due to small numbers. The remaining 65 patients were included in the final study group. The whole group counted 62 cases of hepatomegaly, 65 cases of liver hyperechogenicity and 66 cases of liver surface unevenness. That resulted in 30, 31 and 30 cases, respectively, in the final study group.

The average values obtained in this investigation quantitative parameters are presented in Table 2. A significant correlation (*p* < 0.05) between liver stiffness and AST, INR parameters as well as AST/ALT, age/PLT, APRI and LOK indexes was demonstrated. The liver stiffness and ALT and PLT parameters presented no significant correlation (*p* > 0.05).

To address the problem of outliers in FIB-4, GUCI and King’s Score scales, an analysis without them was also performed and noteworthy correlations were found (*p* < 0.05). The correlation strength (r) is shown in Table 3.

Using the stepwise and backward regression methods, statistically significant models predicting stiffness and SWM1-Vs on the basis of biochemical indexes were created with regression methods. The King’s Score, GUCI, LOK index and FIB-4-based ones were the finest. The obtained models presented maximum R^2^ of 0.43 with *p* < 0.001.

Elevated iQR (%) (*p* = 0.01) and fewer PSWSMs (*p* = 0.05) were found for male patients. Thicker soft tissue occurred in patients with hepatomegaly (*p* = 0.02). Higher liver stiffness (*p* < 0.01) and SWM1-Vs values (*p* < 0.01) were observed in patients with hepatomegaly. In the hepatomegaly group, the VsN (%) (*p* = 0.01) and iQR (%) (*p* = 0.02) were significantly lower. Patients with hyperechoic liver had thicker soft tissue (*p* = 0.01) as well as higher liver stiffness (*p* = 0.03) and SWM1-Vs values (*p* = 0.03). Patients with uneven liver surface presented older age (*p* < 0.01), thicker soft tissues (*p* < 0.01), greater liver stiffness (*p* = 0.01) and SWM1-Vs values (*p* = 0.01). In the group with an uneven liver capsule, a significantly lower VsN (%) (*p* < 0.01) and iQR (%) (*p* < 0.01), as well as a higher number of PSWSMs (*p* < 0.01) were found.

A significant negative correlation was demonstrated for liver stiffness (r = −0.23), SWM1-Vs (r = −0.23), VsN (%) (r = −0.62), iQR (%) (r = −0.66) and soft tissue thickness (*p* < 0.05). The VsN (%) and iQR (%) parameters showed no differences in regards to the PSWSMs in different disease groups.

## 4. Discussion

Up to date studies present a positive correlation of SWE and the biopsy results in cases of chronic viral hepatitis B and C, NAFLD and alcohol liver disease. However, the elastography diagnostic value is still undetermined in autoimmune liver diseases [2,3,6]. Studies addressing the problem of elastography and routinely-used laboratory tests accordance are few, contradictory and refer mostly to transient elastography (TE) [3].

Liver stiffness and SWM1-Vs presented positive correlation with two biochemical parameters (AST and INR) and fibrotest panels, with the King’s Score (r = 0.65) being the strongest one. The literature on the pSWE is limited, but Bota et al. [7] obtained similar results.

Regarding singular biochemical parameters, only INR showed a positive correlation with liver stiffness and the SWM1-Vs parameter. Unlike AST or ALT, INR is not a parameter that can lead to overestimation of the elastographic measurement values [3,8,9,10].

Stepwise regression models were statistically significant, with a maximum R^2^ of 0.42. Therefore, it can be concluded that the application of the above indexes to predict the result of the elastographic test, and thus determine the liver stiffness, is burdened with a significant error and cannot replace the SWE.

Significantly higher stiffness and SWM1-Vs were observed in the groups of patients with hepatomegaly, hyperechogenicity and uneven liver surface. The literature on the effect of a fatty liver on the elastographic measurements are contradictory. Petta et al. [11] claim that significant steatosis should be considered as an independent fibrosis overmeasurement risk factor in TE in NAFLD. Macaluso et al. [12] presented the same association in patients with hepatitis C. In contrast, Wong et al. [13] ruled out the impact of steatosis on elastography in NAFLD.

Soft tissue thickness was negatively but weakly correlated with stiffness and SWM1-Vs. Therefore, it is difficult to consider it as relevant from a practical point of view.

Factors lowering the percentage of proper PSWSMs in regard to VsN (%) are hepatomegaly, hepatic surface nodularity and large soft tissue thickness. Excitation wave energy loss in soft tissue is responsible for VsN (%) drop [3]. The VsN (%) parameter was analysed in correlation with soft tissue thickness in patients with and without hepatomegaly and liver unevenness. Obtained results suggest that an uneven liver surface affects VsN (%), but an increased liver size does not.

In our work, female gender, hepatomegaly, an uneven liver surface and the presence of thick subcutaneous soft tissue reduced the percentage of proper in regard to iQR (%) PSWSMs. A correlation between gender and hepatomegaly could imply that the reduction in iQR (%) in women was due to a higher incidence of an enlarged liver. To obtain a more detailed analysis comparing iQR (%) with gender, we divided the group into subgroups with and without hepatomegaly, and we found that female gender was an independent factor that decreased the PSWSM percentage with the correct iQR/M parameter.

A significant limitation of this study was the small size of the group due to a large number of patients with outdated biochemical tests. To determine the exact relationship between biochemical tests and elastography, it would be necessary to perform them on the same day. Unfortunately, this condition could not be met because this was a retrospective study of outpatients. An additional limitation was the diversity of aetiological factors for liver diseases, which was responsible for the heterogeneity of the investigated population.

## 5. Conclusions

Results of pSWE correlated positively with liver fibrosis biochemical indexes. The King’s Score index presented the strongest correlation. However, prediction of elastographic results cannot be predicted on the basis of individual indexes or panels comprised of several indexes.

A greater soft tissue thickness, uneven liver surface, presence of hepatomegaly and female gender were factors that significantly impaired the pSWE quality indicators. Technical problems encountered during pSWE elastography do not result from any specific liver pathology.

Even if there is a correlation on a group basis for some of the fibrosis scores by biochemistry and SWE, pSWE seems to provide individual and different information on the status of liver fibrosis, which probably makes it a better tool than any fibrosis score.

## Figures and Tables

**Table 1 medicina-57-00040-t001:** Point ranges for the age/PLT index.

Age (years)	Points	PLT (10^9^/L)	Points
<30	0	≥225	0
<40	1	<225	1
<50	2	<200	2
<60	3	<175	3
<70	4	<150	4
≥70	5	<125	5

Age—patients age in years, PLT—platelet count (10^9^/L), Points—number of points obtained.

**Table 2 medicina-57-00040-t002:** Quantitative parameters obtained in the study.

Parameter	All	Female	Male
Stiffness (kPa)	7.64 ± 5.09	6.75 ± 3.13	8.51 ± 6.37
SWM1-Vs (m/s)	1.53 ± 0.46	1.46 ± 0.33	1.59 ± 0.55
VsN (%)	69.78 ± 38.22	68.02 ± 38.35	71.51 ± 38.30
iQR (%)	52.98 ± 32.41	45.11 ± 32.16	60.74 ± 30.96
PSWSM (n)	18.58 ± 8.03	19.77 ± 8.93	17.40 ± 6.89
Liver size (mm)	130.78 ± 23.01	122.58 ± 17.79	138.86 ± 24.77
Soft tissue thickness (mm)	18.45 ± 4.68	18.67 ± 4.80	18.23 ± 4.59
AST/ALT	1.14 ± 0.63	1.23 ± 0.73	1.07 ± 0.53
Age/PLT	4.34 ± 2.74	4.59 ± 2.34	4.11 ± 3.08
APRI index	0.92 ± 0.70	0.97 ± 0.71	0.87 ± 0.70
LOK index	0.45 ± 0.30	0.44 ± 0.30	0.47 ± 0.30
FIB-4 index	2.47 ± 2.59	2.57 ± 2.88	2.38 ± 2.32
King’s score	19.60 ± 20.00	19.94 ± 21.73	19.28 ± 18.51
GUCI	1.09 ± 0.95	1.15 ± 1.05	1.03 ± 0.86

SWM1-Vs—shear wave propagation velocity measurement, VsN—reliability indicator, iQR—interquartile range, PSWSM—number of point shear wave speed measurements, AST—aspartate aminotransferase level, ALT—alanine transaminase level, PLT—platelet count, APRI—AST to platelet ratio index, FIB-4—Fibrosis-4 score, GUCI—Goteborg University cirrhosis index.

**Table 3 medicina-57-00040-t003:** Correlation strength (r) of stiffness and SWM1-Vs parameters with biochemical indexes.

Parameter	AST	INR	Age/PLT	APRI	AST/ALT	LOK index	FIB-4	GUCI	King’s Score
Stiffness (kPa)	0.41 *p* = 0.01	0.54 *p* < 0.001	0.38 *p* = 0.01	0.50 *p* < 0.001	0.44 *p* < 0.001	0.52 *p* < 0.001	0.54 *p* < 0.001	0.59 *p* < 0.001	0.65 *p* < 0.001
SWM1-Vs (m/s)	0.43 *p* < 0.001	0.51 *p* < 0.001	0.40 *p* = 0.001	0.53 *p* < 0.001	0.47 *p* < 0.001	0.54 *p* < 0.001	0.55 *p* < 0.001	0.61 *p* < 0.001	0.65 *p* < 001

## Data Availability

The data presented in this study are available on request from the corresponding author.

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
