# Peer review of "Analysis of Point Shear Wave Elastography and Biochemical Markers for the Detection of Liver Fibrosis"

_medicina, 2021, doi:10.3390/medicina57010040_

Round 1

Reviewer 1 Report

In the manuscript " Analysis of  point shear wave elastography and biochemical markers for the detection of liver fibrosis", Maciej  Cebula and colleagues performed a retrospective study in order to better evaluate the role of point shear wave elastography in liver fibrosis.  The paper could have a clinical significance. However, some major  criticisms are present, as follows:

- In line 55-57  the authors state that “patient records were obtained in the form of anonymised reports, and the identification of individual patients solely based on the collected data is impossible”. Please explain this phrase and explain how did you gather your data base and the biochemical values if you cannot identify the patients.

- Please explain why and how did you choose the indexed mentioned in the paper and their importance in clinical practice. This makes the results hard to reproduce.

- In lines 132-146 the authors have compared pSWE from different liver disease, however the study lot is too varied and they compare even liver surface with uneven liver surface, which for this lot size is not accurate.

- There is a lack of novelty to the paper and it has low reproducibility.

-Conclusions should be better stated in order to better highlight the point of the study.

- Minor English revision.

Author Response

Dear Sir or Madam,

I would like to thank you for your insight regarding our manuscript and apologise for a late response that was caused by my illness.

- In line 55-57  the authors state that “patient records were obtained in the form of anonymised reports, and the identification of individual patients solely based on the collected data is impossible”. Please explain this phrase and explain how did you gather your data base and the biochemical values if you cannot identify the patients.

Our study is a retrospective one. We extracted all the data from hospital and private clinic databases in form of anonymised spreadsheets. To be more specific, we asked our informaticians to make a report from databases for all the patients that underwent the elastography in certain time period and we obtained an excel spreadsheet that contained internal database ID number, ICD 10 code, sex, age, laboratory results and pSWE report. As the internal database ID number allows identification of patient only by administrators of the system, we were unable to see the patients credentials and database was treated as anonymised.

- Please explain why and how did you choose the indexed mentioned in the paper and their importance in clinical practice. This makes the results hard to reproduce.

We had to compromise in that regard, as retrospective character of the study limited us heavily. We checked what laboratory results are available for the analysed group of patients and on this basis we listed the indices we were able to calculate. To minimize the time bias we narrowed the lab results to up to 7 days before or after the elastography examination. The indices were evaluated and picked by experienced gastroenterologist, Prof. M. Hartleb, Head of Department of Gastroenterology and Hepatology.  

- In lines 132-146 the authors have compared pSWE from different liver disease, however the study lot is too varied and they compare even liver surface with uneven liver surface, which for this lot size is not accurate.

We are aware that the study is limited due to the group size, but we are unable to elevate it significantly in foreseeable time. Due to that we decided to report results at this stage and focus on more specific, prospective studies regarding the elastography method.

- There is a lack of novelty to the paper and it has low reproducibility.

The study confirms up to date knowledge in the field. The novelty in our opinion is comparative nature of the study in regard of analysed biochemical indices. To be able to explore the problem further in our opinion a multicentre, prospective study would be required in order to gather large enough group size. As practical appliance of potentially obtained results seems to be minimal and pSWE is slowly replaced by the 2D-SWE, it seems pointless at this point.

-Conclusions should be better stated in order to better highlight the point of the study.

We edited the conclusions according to suggestion.

- Minor English revision.

We are not native English speakers, so prior to submission the manuscript was sent to professional editor for revision. At this point it is hard for us to pick the imperfections, especially without more specific indication. If there are any mistakes overlooked they will most probably be corrected in the final revision prior to the publication. 

We tried our best to address all the issues you pointed. We hope that our response is up to your standards and you will be willing to allow us to publish our results. Regardless of the outcome we would like to thank you for the dedicated time and constructive critic of our paper.

Reviewer 2 Report

This is quite a confusing paper! This paper tries to compare pSWE and biological tests, but without a reference method. Small sample and heterogenous!

Which is the feasibility of pSWE (how many patients did not have IQR/M≤30%)?

What means proper preparation of patients for pSWE (fasting)?

There is no real introduction of the paper!

What means patients with hepatomegaly, where you made a lot of correlation? What means liver size? Which is the value of liver size in practice?

The only merit of this paper is that found that King’s score is best correlated with pSWE!

Author Response

Dear Sir or Madam,

I would like to thank you for your insight regarding our manuscript and apologise for a late response that was caused by my illness.

This is quite a confusing paper! This paper tries to compare pSWE and biological tests, but without a reference method. Small sample and heterogenous!

We are aware that the study is limited due to the group size, but we are unable to elevate it significantly in foreseeable time, especially in regard of rare diseases. Due to that we decided to report results at this stage and focus on more specific, prospective studies regarding the elastography method. The reference method would have to be a liver biopsy. Due to decreasing amount of biopsies performed in our centres and retrospective character of the study we were simply unable to gather large enough group of patients to be able to compare the method.

Which is the feasibility of pSWE (how many patients did not have IQR/M≤30%)?

The used USG system shows the IQR/M for each of the measurements as well as the whole examination (10 measurements). We presented the results as % of point measurements with IQR/M<30% in table 2. To answer the question in regard of whole study group, IQR/M<30% for the whole examination was not observed in 37 out of 139 cases.

What means proper preparation of patients for pSWE (fasting)?

The proper preparation is defined by EFSUMB guidelines (mentioned in lines 48-49, citation 3). The EFSUMB guidelines define the preparation in recommendation 6 “ Patients should fast for a minimum of 2 hours and rest for a minimum of 10 minutes before undergoing liver stiffness measurement with SWE”

There is no real introduction of the paper!

We are sorry but due to lack of specificity and constructive aspect of this remark we do not know how to address it. The introduction of the paper have been modified according to other reviewers suggestions, so we hope that it will be satisfactory.

What means patients with hepatomegaly, where you made a lot of correlation? What means liver size? Which is the value of liver size in practice?

The methodology of liver size measurement is defined in PTU guidelines (mentioned in line 49, citation 4). The PTU guidelines define the liver size measurement as “(…) liver measurement is performed in a standard way by placing the ultrasound head in the right parasternal line at the epigastric level, obtaining cross-section showing drainage of the hepatic veins and all hepatic veins. In such a projection, the size of the liver is measured in the mid-clavicular line.” (fragment translated from polish). The established norm for this measurement is up to 130mm. Patients with measurement >130mm were classified as patients with hepatomegaly.

The only merit of this paper is that found that King’s score is best correlated with pSWE!

We are glad that you were able to see some positive aspect of the study.

We tried our best to address all the issues you pointed. We hope that our response is up to your standards and you will be willing to allow us to publish our results. Regardless of the outcome we would like to thank you for the dedicated time and constructive critic of our paper.

Reviewer 3 Report

In this paper, the authors make correlations and perform regression analyses  to link single parameters and known panels of biochemical parameters to the  result of Shear wave based elastrography providing shear-wave speed measurements (Vs) as well as elasticity (E) values calculated on the basis of  Vs. It may be interesting to explore if liver stiffness can be determined based on the formerly known (insufficient) panels predicting the status of liver fibrosis, or if SWE measurments represent an individual and more representative and sophisticated examination method to monitor liver fibrosis. Their  conclusion is that none of these indices allow a reliable prodiction of elastographic test results, even if  a significant correlation exists on group basis for some of them. 

SWE exists in several versions depending on the  US manufacturer. Hitachi is  one of the pioneers in the market and was one of the first makers to  provide  strain elastography, and more recently SWE. SWE as such is quite similar in technology between the US manufacturers, particularly point SWE which do not provide a colour map or any information of the quality of the  measurement before the measurement, but provides a percentage of the  velocity measuremnts that fall within an expected range of velovities  set up by the manufacturer. This is provided as a quality percentage on the screen, the VsN. SWE has previously been compared to fibrosis in histology and in several studies with transient elastography. SWE represent a quick and non- invasive method for assessing the liver fibrosis, but depends on adequate  B-mode imaging, temporary  breath hold, fasting condition, no AST flares and some other conditions. 

Major Comments:

  1. At the end of the introduction the authors must state the primary aim and therafter any secondary aims of the paper, there is no clear  description of  the  motivation of performing this study.
  2. The authors must  define cleraly  what are the outcome parameters, and what do they  define as the quality parameters. This is also the  case for  the fibrosis  biomarkers. 
  3. The authors  introduce  a new experession: "Measurement correctness parameters" and refer to  "obtaining  the  predefined  levels of these measurments", but they  do not state  clerly  which  parameters this is. We assume that they  refer to VsN >60% and IQR or  IQR/median < 30% for  Elasticity and <15 % for SWE velocity measurements. But this needs to  be written. 
  4. The  study  seems to include any patient with an elevated AST or ALT level  no matter the diagnosis, background or duration. This creates a very unhomogenic patient cohort. You also argu to exclude  two  specific patients due to a seldom specific  diagnosis, Haemochromatosis and benign recurrent intraheptic  cjholestasis, givenn the  wide inclusion criteria it makes no sense to exclude these two patients. 
  5. Under limitations the fact that two or more surrogate parameters for fibrosis are compared and some correlation is found, without a histological reference is a major limitation for knowing which method is better. 
  6. The authors  find through regression analysis that the more solid  B-mode  signs of liver cirrosis, such as a  bulky capsule, affects VsN value which is experienced with any elastography method, and is based on more uneven elasticity in advanced fibrosis leading to more dispersion and less measurments falling into the range of expected values. This has some value, but only confirms what is already experienced. The B-mode image is  specific, but not  sensitive, as advanced fibrosis can be present without the  finding of  a bulky capsuel, this  is  where  SWE comes in as a very useful parameter.  
  7. In the conclusion they fail to conclude that even if there is a correlation on group basis for some of the fibrosis scores by biochemistry and SWE, pSWE seems to provide individual and different information on the status of liver fibrosis which probably makes it a better tool than any  fibrosis score.

Minor comments:

  1. "pSWE correctness parameters" is a misnomer and should be exchanged with  "pSWE quality  indicators" or  "indicator parameters". Their  proposed function and recomended values should be stated. 
  2. The  nature of  VsN should  be presented more in detail, and not as a  "black box" quality parameter. Hitachi has information on its home pages. 
  3. The Institutional Ethical  committee probably has a an official  name in English, please use this in the  M&M section.
  4. P2, L 68-75 refers data that  belongs in the  result  section, not in the  M&M section.
  5. P2: 77. What is  the "soft tissue  thickness", is it the subcutaneous tussue over the capsule? Please be more precise. Did you have a lower cut-off to define it as "thick" or did you use  a numerical (mm) scale?
  6. P2, L78: What is meant by "high liver orientation"?
  7. Table 1: Please provide explanation to  the  abbreviations as a table-legend, tables  should be readable  on their own. 
  8. Table 3: Why do you make a correlation with both the Elasticity value (kPa) and the Vs (m/s) as E is  calculated from Vs? The correlation, r is  higher for Vs than for E, for all correlations have you got any explanation for this? Is there a significant difference between E and Vs correlations?
  9. P4, L 130: You  specifically mention Kings´s score, LOK and FIB-4 as best correlations, but from the  table it  seems that  Guci is also among the  best correlated scores, why is  it  left out?
  10. Is it  really necessary  to include seven different fibrosis scoes and 2 individual biochemical parameters? Some of  them are closesely related to  each other, such as GUCI and King´s score.
  11. Most of the  correlations have r values from 0.50-0.65. Were there any  significant difference
  12. Table 4 consits of one line and could be omitted and just  written in the  text. 
  13. P5, L 163: "Unlike  AST or ALT, INR is not a parameter that can potentially increase elastographic measurement values" What is the rationale  behind this sentence? Is it the E value that should be increased or the quality measurement parameters (VSN and IQR)? INR is  known as one of the most robust indicators of  advanced liver disease reflecting inadequacy of  liver synthesis of  coagulation factors, while  AST and ALT in this  phase pseudo- normalize. This sentence makes no sense to me. 
  14. In limitations, yoy  refer to "outtdated biochemical tests".What  does this mean, did you  have some upper limitation for  how long time between the  pSWE and the  biochemistry? The main limitation of  the  study  is  the  diverse etiology and the  lack of  a gold standard, e.g.: histology. 
  15. Other references that might be of interest: Novel serum and bile protein markers predict primary sclerosing cholangitis disease severity and prognosis. Vesterhus M, Holm A, Hov JR, Nygård S, Schrumpf E, Melum E, Thorbjørnsen LW, Paulsen V, Lundin K, Dale I, Gilja OH, Zweers SJLB, Vatn M, Schaap FG, Jansen PLM, Ueland T, Røsjø H, Moum B, Ponsioen CY, Boberg KM, Färkkilä M, Karlsen TH, Lund-Johansen F.J Hepatol. 2017 Jun;66(6):1214-1222. doi: 10.1016/j.jhep.2017.01.019. Epub 2017 Feb 2.PMID: 28161472   Ultrasound and Point Shear Wave Elastography in Livers of Patients with Primary Sclerosing Cholangitis. Mjelle AB, Mulabecirovic A, Hausken T, Havre RF, Gilja OH, Vesterhus M.Ultrasound Med Biol. 2016 Sep;42(9):2146-55. doi: 10.1016/j.ultrasmedbio.2016.04.016. Epub 2016 Jun 2.PMID: 27262519  

Author Response

Dear Sir or Madam,

I would like to thank you for your insight regarding our manuscript and apologise for a late response that was caused by my illness.

I would like to mention one problem in the review in general – there seems to be mismatch of pages/lines mentioned by reviewer and paper draft we received from the MDPI. We do not know the reason, but tried our best to find mentioned fragments and address the issues. We hope that it will not impact negatively the final decision of the reviewer.

In this paper, the authors make correlations and perform regression analyses  to link single parameters and known panels of biochemical parameters to the  result of Shear wave based elastrography providing shear-wave speed measurements (Vs) as well as elasticity (E) values calculated on the basis of  Vs. It may be interesting to explore if liver stiffness can be determined based on the formerly known (insufficient) panels predicting the status of liver fibrosis, or if SWE measurments represent an individual and more representative and sophisticated examination method to monitor liver fibrosis. Their  conclusion is that none of these indices allow a reliable prodiction of elastographic test results, even if  a significant correlation exists on group basis for some of them. 

SWE exists in several versions depending on the  US manufacturer. Hitachi is  one of the pioneers in the market and was one of the first makers to  provide  strain elastography, and more recently SWE. SWE as such is quite similar in technology between the US manufacturers, particularly point SWE which do not provide a colour map or any information of the quality of the  measurement before the measurement, but provides a percentage of the  velocity measuremnts that fall within an expected range of velovities  set up by the manufacturer. This is provided as a quality percentage on the screen, the VsN. SWE has previously been compared to fibrosis in histology and in several studies with transient elastography. SWE represent a quick and non- invasive method for assessing the liver fibrosis, but depends on adequate  B-mode imaging, temporary  breath hold, fasting condition, no AST flares and some other conditions. 

The above fragment of review shows reviewers expertise in the field and deep understanding of the topic. We would like to underline it, as the quality of review here is very high and we are grateful for dedicated time and all the insight.

Major Comments:

  1. At the end of the introduction the authors must state the primary aim and therafter any secondary aims of the paper, there is no clear  description of  the  motivation of performing this study.

The paper have been modified accordingly.

  1. The authors must  define cleraly  what are the outcome parameters, and what do they  define as the quality parameters. This is also the  case for  the fibrosis  biomarkers. 

The paper have been modified to better visualise what was the role of each of the parameters used in the study. We hope that it satisfies the reviewer.

  1. The authors  introduce  a new experession: "Measurement correctness parameters" and refer to  "obtaining  the  predefined  levels of these measurments", but they  do not state  clerly  which  parameters this is. We assume that they  refer to VsN >60% and IQR or  IQR/median < 30% for  Elasticity and <15 % for SWE velocity measurements. But this needs to  be written. 

The reviewer listed all the parameters that were taken in the consideration. While writing the paper we addressed this issue in line 48-49 referring to the guidelines. As it seems to be not enough we added the paragraph explaining the matter.

  1. The  study  seems to include any patient with an elevated AST or ALT level  no matter the diagnosis, background or duration. This creates a very unhomogenic patient cohort. You also argu to exclude  two  specific patients due to a seldom specific  diagnosis, Haemochromatosis and benign recurrent intraheptic  cjholestasis, givenn the  wide inclusion criteria it makes no sense to exclude these two patients. 

In previously received review as well as in feedback from senior staff I received the opposite suggestion to remove the singular cases. It was argued that being not representative they only add the noise to the analysis. I agree that the study sample is small and heterogenous but unfortunately we are unable to elevate it significantly in foreseeable time. Due to that we decided to report results at this stage and focus on more specific, prospective studies regarding the elastography method.

  1. Under limitations the fact that two or more surrogate parameters for fibrosis are compared and some correlation is found, without a histological reference is a major limitation for knowing which method is better. 

The reference method would have to be a liver biopsy. Due to decreasing amount of biopsies performed in our centres and retrospective character of the study we were simply unable to gather large enough group of patients to be able to compare the methods.

  1. The authors  find through regression analysis that the more solid  B-mode  signs of liver cirrosis, such as a  bulky capsule, affects VsN value which is experienced with any elastography method, and is based on more uneven elasticity in advanced fibrosis leading to more dispersion and less measurments falling into the range of expected values. This has some value, but only confirms what is already experienced. The B-mode image is  specific, but not  sensitive, as advanced fibrosis can be present without the  finding of  a bulky capsuel, this  is  where  SWE comes in as a very useful parameter.  

We are glad that reviewer find some value in our study. We encountered the problem with obtaining the proper measurements in advanced fibrosis in everyday practice but were not sure if the problem is really there or is it coincidental observation. Due to that we decided to search for the visible in B-mode factors that could be a determinant of observed effect as a secondary aim of the study.   

  1. In the conclusion they fail to conclude that even if there is a correlation on group basis for some of the fibrosis scores by biochemistry and SWE, pSWE seems to provide individual and different information on the status of liver fibrosis which probably makes it a better tool than any  fibrosis score.

Conclusion added.

Minor comments:

  1. "pSWE correctness parameters" is a misnomer and should be exchanged with  "pSWE quality  indicators" or  "indicator parameters". Their  proposed function and recomended values should be stated

Changed according to the comment.

  1. The  nature of  VsN should  be presented more in detail, and not as a  "black box" quality parameter. Hitachi has information on its home pages. 

The VsN is defined on Hitachi web page (https://www.hitachi.com/businesses/healthcare/products-support/contents/us-tech/elastography/index.html) as follows:

“Reliability indicator VsN

Multiple Vs value sets are acquired at a time, and the percentage of effective values in them is displayed as a reliability indicator VsN. This function allows you to test the validity of the measurement.”

In our paper in lines 78-79 of new draft VsN is defined as “(…) percentage of effective Vs efficiency rate (VsN, reliability indicator that allows to test the measurement validity)(…)”.

We hope that this change satisfies reviewer.

  1. The Institutional Ethical  committee probably has a an official  name in English, please use this in the  M&M section.

The Ethical committee lacks the official English name – double checked this information with the University representatives. In such situation we left the official Polish name instead of translating it on our own.

  1. P2, L 68-75 refers data that  belongs in the  result  section, not in the  M&M section.

Changed according to the comment.

  1. P2: 77. What is  the "soft tissue  thickness", is it the subcutaneous tussue over the capsule? Please be more precise. Did you have a lower cut-off to define it as "thick" or did you use  a numerical (mm) scale?

Soft tissue thickness was measured as the distance between the liver capsule and the probe surface on each of the point measurements. The value was in numerical (mm) scale. Information added.

  1. P2, L78: What is meant by "high liver orientation"?

High liver orientation is defined by PTU guidelines (reference 4). The position of the liver is assessed by applying the ultrasound probe in the V intercostal space in the anterior and middle axillary line. If the location of the upper border of the liver is observed above the 5th intercostal space, the location is determined as high.

  1. Table 1: Please provide explanation to  the  abbreviations as a table-legend, tables  should be readable  on their own. 

Changed according to the comment.

  1. Table 3: Why do you make a correlation with both the Elasticity value (kPa) and the Vs (m/s) as E is  calculated from Vs? The correlation, r is  higher for Vs than for E, for all correlations have you got any explanation for this? Is there a significant difference between E and Vs correlations?

The only difference here is the accuracy of the measurement presented by the system. Our USG report elasticity with two and Vs with one decimal, so the difference is only due to the rounding. Unfortunately we are unable to unify it retrospectively as saved examination reports do not contain the raw data. We showed elasticity and Vs to prove that outcomes are not a result of this rounding problem. The Vs and E of course do not show significant differences and r difference is also not significant.

  1. P4, L 130: You  specifically mention Kings´s score, LOK and FIB-4 as best correlations, but from the  table it  seems that  Guci is also among the  best correlated scores, why is  it  left out?

Yes, you are right, it was left out. To be frank I do not know why, most probably got deleted by mistake in one of the corrections. Guci added.

  1. Is it  really necessary  to include seven different fibrosis scoes and 2 individual biochemical parameters? Some of  them are closesely related to  each other, such as GUCI and King´s score.

The indices were evaluated and picked by experienced gastroenterologist, Prof. M. Hartleb, Head of Department of Gastroenterology and Hepatology. To my knowledge the aim was to compare as many of them as possible and show “the best one”.

  1. Most of the  correlations have r values from 0.50-0.65. Were there any  significant difference

From the statistical point of view presented correlation values are not significantly different. That can be a result of low sample size. The r values go from 0,38 to 0,65, so what is interesting in our opinion is that they are placed in two partitions of the Guilford scale, despite of their close relation, as you mentioned.

  1. Table 4 consits of one line and could be omitted and just  written in the  text. 

Table deleted, information included in the text.

  1. P5, L 163: "Unlike  AST or ALT, INR is not a parameter that can potentially increase elastographic measurement values" What is the rationale  behind this sentence? Is it the E value that should be increased or the quality measurement parameters (VSN and IQR)? INR is  known as one of the most robust indicators of  advanced liver disease reflecting inadequacy of  liver synthesis of  coagulation factors, while  AST and ALT in this  phase pseudo- normalize. This sentence makes no sense to me. 

What we wanted to tell here is that it is proven that elevated AST and/or ALT can lead to overestimation of liver fibrosis in elastography. AST and ALT are listed in EFSUMB guidelines in recommendation 7 as potential confounding factors with above effect. To our knowledge INR was not proven to have such effect and we were unable to find papers proving the same effect for the INR alone. The INR is also not mentioned in the EFSUMB guidelines.

  1. In limitations, yoy  refer to "outtdated biochemical tests".What  does this mean, did you  have some upper limitation for  how long time between the  pSWE and the  biochemistry? The main limitation of  the  study  is  the  diverse etiology and the  lack of  a gold standard, e.g.: histology.

Yes, there was a time limitation between pSWE and biochemistry, ± 7 days to be exact. It is mentioned in P2, L 66-68. We placed it to avoid the issue of pSWE bias due to acute inflammation at the time of examination, but not yet/still being visible in the biochemistry. Changed the sentence to better show this information. I agree about the rest of listed limitations, but unfortunately due to the character of the study we were unable to address them better.

  1. Other references that might be of interest: Novel serum and bile protein markers predict primary sclerosing cholangitis disease severity and prognosis. Vesterhus M, Holm A, Hov JR, Nygård S, Schrumpf E, Melum E, Thorbjørnsen LW, Paulsen V, Lundin K, Dale I, Gilja OH, Zweers SJLB, Vatn M, Schaap FG, Jansen PLM, Ueland T, Røsjø H, Moum B, Ponsioen CY, Boberg KM, Färkkilä M, Karlsen TH, Lund-Johansen F.J Hepatol. 2017 Jun;66(6):1214-1222. doi: 10.1016/j.jhep.2017.01.019. Epub 2017 Feb 2.PMID: 28161472   Ultrasound and Point Shear Wave Elastography in Livers of Patients with Primary Sclerosing Cholangitis. Mjelle AB, Mulabecirovic A, Hausken T, Havre RF, Gilja OH, Vesterhus M.Ultrasound Med Biol. 2016 Sep;42(9):2146-55. doi: 10.1016/j.ultrasmedbio.2016.04.016. Epub 2016 Jun 2.PMID: 27262519  

Thank you for the references.

The first one is an interesting study on PSC patients but focuses on the proteins panels, where most of them are not used in everyday practice, so such results are not available to us. Despite the quality of the study and promising results its practical appliances in our setting are minimal due to practical, mostly economic reasons. We are also unable to relate this work to our paper for the obvious reason of lack of the data on our side.

The second one focuses on the pSWE in PSC with or without accompanying AIH. It has much larger and homogenous group of patients and addresses all the issues our paper suffers from.  We are currently preparing a prospective study and this paper will be a valuable addition to the references. I would only like to point out, that in this paper the pSWE measurements were made in right and left liver lobes. The pSWE  of the left liver lobe is still a discussed issue, as part of the authors claim that heart beat can influence the elastographic measurement. Up to my knowledge it was proven that there is a notable, but statistically insignificant difference between right and left lobe (segments IVA and IV B) measurements (seen the paper presented at ECR 2019, unfortunately do not have the reference) that is also confirmed here. To completely avoid this issue an MRE seems to be a reasonable tool to assess the liver stiffness in advanced studies and probably should be added to such investigations in future.

We tried our best to address all the issues you pointed. We hope that our response is up to your standards and you will be willing to allow us to publish our results. Regardless of the outcome we would like to thank you for the dedicated time and constructive critic of our paper.

Round 2

Reviewer 1 Report

Accept in present form

Author Response

Thank you for the acceptance of our work.

We resubmitted slightly modified manuscript in accordance to other reviewers feedback.

Reviewer 3 Report

The authors have answered or explained according to my comments and made changes to several of the commented subjects. The resubmitted manuscript is improved most particularly the  aims are cleraly stated and the conclusion is  supported by the  findings, but some minor aspects still needs to be addressed. 

Minor  comments: 

  1. P 3, L112-114 "In search of technically difficult in the elastographic examination liver pathologies, a retrospective analysis was conducted, and the number of point shear wave speed measurements (PSWSMs) needed to obtain a sound result for the entire study was counted." This  sentence is unclear and needs to be rephrased. 
  2. P 5 L173-175: "Although this relationship was statistically significant, building the liver fibrosis diagnosis based only on the state of the extrinsic coagulation system seems tenuous. Unlike AST or ALT, INR is not a parameter that can potentially increase elastographic measurement values." INR is one of  the  most reliable parameters indicating loss of liver synthesis in advanced liver disease. In the abscence of  anticoagulation therapy, it correlates  well with the presence of advanced fibrosis or cirrhosis, and it is therefore more expected than ALT or AST which may be caused by other sources or be transient even in quite healthy livers. I suggest to remove or modify this sentence. 
  3. Table 3: The authors are aware that SW velocity in m/s is the measured parameter and that Stiffness in kPa is a deduction based on: C x (Vs)2? In this deduction, (C is  tissue  density) + the tissue is regarded as incompressible (Poisson`s ratio: 0.5) and the US speed as constant in tissues. For most of the biochemical parameters the correlation ratio (r) is slighly higher for SW velocities than for Elasticicty (E), even if the difference probably is not significant. This is also an interesting  finding that  could have been addressed in the  discussion. As long as this  remains uncommented it seems to me as if the authors are unaware of this close relationship and present the Elasticity (kPa) and SWVs (m/s) as two separate  parameters. Table 3 also looks a bit incomplete without noting the p-values of the calculated r-values. 
  4. Language would in general benefit form native English editing.

Author Response

Dear Sir of Madame,

The authors have answered or explained according to my comments and made changes to several of the commented subjects. The resubmitted manuscript is improved most particularly the  aims are cleraly stated and the conclusion is  supported by the  findings, but some minor aspects still needs to be addressed. 

Thank you for the positive evaluation of our work on the manuscript.

Minor  comments: 

P 3, L112-114 "In search of technically difficult in the elastographic examination liver pathologies, a retrospective analysis was conducted, and the number of point shear wave speed measurements (PSWSMs) needed to obtain a sound result for the entire study was counted." This  sentence is unclear and needs to be rephrased. 

Sentence rephrased.

P 5 L173-175: "Although this relationship was statistically significant, building the liver fibrosis diagnosis based only on the state of the extrinsic coagulation system seems tenuous. Unlike AST or ALT, INR is not a parameter that can potentially increase elastographic measurement values." INR is one of  the  most reliable parameters indicating loss of liver synthesis in advanced liver disease. In the abscence of  anticoagulation therapy, it correlates  well with the presence of advanced fibrosis or cirrhosis, and it is therefore more expected than ALT or AST which may be caused by other sources or be transient even in quite healthy livers. I suggest to remove or modify this sentence. 

We of course agree, this part was meant to adress overmeasurement risk connected with elevated ALT/AST. First sentence removed, second modified.

Table 3: The authors are aware that SW velocity in m/s is the measured parameter and that Stiffness in kPa is a deduction based on: C x (Vs)2? In this deduction, (C is  tissue  density) + the tissue is regarded as incompressible (Poisson`s ratio: 0.5) and the US speed as constant in tissues. For most of the biochemical parameters the correlation ratio (r) is slighly higher for SW velocities than for Elasticicty (E), even if the difference probably is not significant. This is also an interesting  finding that  could have been addressed in the  discussion. As long as this  remains uncommented it seems to me as if the authors are unaware of this close relationship and present the Elasticity (kPa) and SWVs (m/s) as two separate  parameters. Table 3 also looks a bit incomplete without noting the p-values of the calculated r-values. 

Yes, we are aware that stiffness is a wave speed recalculated with the use of the Young modulus and that some imperfect assumptions are made in the process. Up to our knowledge this mathematical modification should not affect the correlation in any significant way (that is actually the basic reason for use of the Stiffness instead of the wave speed). Addressing the slightly higher correlations for SWV at this point is possible only in form of theoretical divagation about the reasons, as we do not have the tissue sample and are unable to actually show the underlaying proof for presented reasoning. As already stated, the difference can be partially due to rounding of the results in reports.

Raporting only one of the above (Elasticity or SWVs) creates problem for part of the readers – while Stiffness is used in everyday practice in Europe and most of the Asia, the SWVs is parameter of choice in USA.

The p values were added to Table 3.

Language would in general benefit form native English editing.

As we already stated the text was edited by native English speaker prior to submission. At this point, as non-native speakers we are unable to correct it on such short notice. There is high probability that corrections will be made before the publication.